Characterization of the bacterial microbiota composition and evolution at different intestinal tract in wild pigs (Sus scrofa ussuricus)

Yang Guangli guangliyang@163.com 1
Shi Chuanxin 1
Zhang Shuhong 1
Liu Yan 2
Li Zhiqiang 1
Gao Fengyi 1
Cui Yanyan 1
Yan Yongfeng 1
Li Ming liming@henau.edu.cn 3
1 Department of Biology and Food Sciences, Shangqiu Normal University , Shangqiu City , Henan Province , China
2 College of Animal Husbandry Engineering, Henan Vocational College of Agricultural , Zhengzhou City , Henan Province , China
3 Engineering College of Animal Husbandry and Veterinary Science, Henan Agricultural University , Zhengzhou City , Henan Province , China
Kumar Abhishek
Electronic publication date: 2020 May 26
Publication date: 2020
Volume: 8
Electronic Location ID: e9124
Received 2019 Oct 9; Accepted 2020 Apr 14
Copyright: ©2020 Yang et al.
Copyright year: 2020
Copyright holder: Yang et al.
License: This is an open access article distributed under the terms of the Creative Commons Attribution License, which permits unrestricted use, distribution, reproduction and adaptation in any medium and for any purpose provided that it is properly attributed. For attribution, the original author(s), title, publication source (PeerJ) and either DOI or URL of the article must be cited.
License URL: https://creativecommons.org/licenses/by/4.0/

Keywords: Wild pigs, Gut microbiota, Structure and composition, 16S rRNA gene, Evolution

Funding: Foundation of He’nan Educational Committee of China 14B230017 Industry University Research Cooperation Project of He’nan Science and Technology Committee of China 182107000041 Science and Technology Opening Cooperation Project of He’nan Science and Technology Committee of China 182106000039 This study was supported by the Foundation of He’nan Educational Committee of China (14B230017), the Industry University Research Cooperation Project of He’nan Science and Technology Committee of China (182107000041), and the Science and Technology Opening Cooperation Project of He’nan Science and Technology Committee of China (182106000039). The funders had no role in study design, data collection and analysis, decision to publish, or preparation of the manuscript.

==============================
Commensal microorganisms are essential to the normal development and function of many aspects of animal biology, including digestion, nutrient absorption, immunological development, behaviors, and evolution. The specific microbial composition and evolution of the intestinal tracts of wild pigs remain poorly characterized. This study therefore sought to assess the composition, distribution, and evolution of the intestinal microbiome of wild pigs. For these analyses, 16S rRNA V3-V4 regions from five gut sections prepared from each of three wild sows were sequenced to detect the microbiome composition. These analyses revealed the presence of 6,513 operational taxonomic units (OTUs) mostly distributed across 17 phyla and 163 genera in these samples, with Firmicutes and Actinobacteria being the most prevalent phyla of microbes present in cecum and jejunum samples, respectively. Moreover, the abundance of Actinobacteria in wild pigs was higher than that in domestic pigs. At the genus level the Bifidobacterium and Allobaculum species of microbes were most abundant in all tested gut sections, with higher relative abundance in wild pigs relative to domestic pigs, indicating that in the process of pig evolution, the intestinal microbes also evolved, and changes in the intestinal microbial diversity could have been one of the evolutionary forces of pigs. Intestinal microbial functional analyses also revealed the microbes present in the small intestine (duodenum, jejunum, and ileum) and large intestine (cecum and colon) of wild pigs to engage distinct metabolic spatial structures and pathways relative to one another. Overall, these results offer unique insights that would help to advance the current understanding of how the intestinal microbes interact with the host and affect the evolution of pigs.

Introduction

The intestines of mammals are colonized by trillions of microorganisms wherein they play a dynamic role in controlling host physiological, immunological, and digestive processes (Brestoff & Artis, 2013). The vertebrate intestinal microbiome plays a key role in regulating host biology, and further research suggests it may also influence vertebrate evolution (Sharpton, 2018). Recent years have witnessed a growing number of studies focusing on the composition of the porcine intestinal microbiota (Crespo-Piazuelo et al., 2018; Gao et al., 2019; Kelly et al., 2017; Xiao et al., 2018; Yang et al., 2016; Zhao et al., 2015), with correlations having been drawn between this composition and pig average daily weight gain (Mach et al., 2015; Ramayo-Caldas et al., 2016), feed conversion efficiency and intake (Camarinha-Silva et al., 2017; McCormack et al., 2017; Quan et al., 2018; Quan et al., 2019). A few studies have focused on the microbial makeup in distinct regions of the intestinal tract. Furthermore, these studies have largely been restricted to certain breeds of domestic pigs such as the Large White (Zhao et al., 2015), Laiwu (Yang et al., 2016), Gloucestershire Old Spot (Kelly et al., 2017), Lberian pigs (Crespo-Piazuelo et al., 2018), Jinhua and Landrace (Xiao et al., 2018), and Chinese Shanxi Black breeds (Gao et al., 2019).

Wild pigs are the most closely-related species to domestic pigs, with the divergence between these two populations having been initiated in Eurasia roughly 10,000 years ago (Larson et al., 2005). Domestic pigs are now distinct from their wild counterparts in various ways, such as phenotype, as evidenced by coat color differences (Yang et al., 2019). The specific microbial composition and evolution of the intestinal tracts of wild pigs, however, remain poorly characterized. Many studies have found that at different evolutionary stages, vertebrate species exhibit distinct patterns of microbiota diversity and functional that are correlated with the evolutionary history of that species (Brooks et al., 2017; Gaulke et al., 2017; Groussin et al., 2017; Ley et al., 2008). There is also evidence to support the fact that microbial signatures may be heritable (Koskella, Hall & Metcalf, 2017), and that there are microbial and host co-phylogenetic patterns owing to their close reliance on one another (Brooks et al., 2017; Gaulke et al., 2017; Groussin et al., 2017; Moeller et al., 2016). These results are consistent with a model wherein vertebrate evolution is influenced by the composition of the gut microbiome (Gaulke et al., 2017; Groussin et al., 2017; Ley et al., 2008; Muegge et al., 2011), with many processes having the potential to drive or shape this relationship (Brooks et al., 2017). To date, uncertainty prevails as to how significantly the gut microbiome diversity affects vertebrate evolution.

Recently, next generation sequencing has provided more 16S rRNA gene sequence reads that can be further analyzed to provide in depth studies microbial populations and compositions, and to facilitate more complex analyses of the link between the microbiome and vertebrate evolution. Efforts to assess the functional capabilities of the microbiota in a given organism have been supported by metagenomics annotation efforts (Nayfach et al., 2015). The studies in rodents have demonstrated that microbiome transplantation can alter feed efficiency in these animals (Brooks et al., 2017), suggesting that certain microbial phyla are optimized to support host growth and survival, thus tying the microbiome directly to key evolutionary processes. When inbred mice maintained in laboratory facilities receive feces transplant from wild mice, improving their resistance, it suggests that the artificial inbreeding might have led to microbiome drift toward a less resilient phylotype (Rosshart et al., 2017). Additional research has identified bacteria capable of degrading oxalate in the gut of rodents, suggesting that these bacteria support the adaptation of these animals to additional dietary niches by expanding their ability to utilize certain nutrient sources (Kohl et al., 2014). The intestinal microbiota of domestic pigs, wild pigs, and red river hog studies reported that the Lactobacilli bacteria are primarily present in domesticated or captured pigs, whereas Bifidobacterium dominates the intestinal microbiota of wild animals (Ushida et al., 2016).

Further studies of the gut microbiome of wild pigs may, therefore, offer valuable insight into the relationship between this microbial community and porcine evolution. As such, in the present study we sampled the gut microbiome in five distinct regions of the gastrointestinal tract of wild pigs (duodenum (DU), jejunum (JE), ileum (IL), cecum (CE), and colon(CO)), using 15 samples for 16S rRNA sequencing to facilitate microbial characterization. Using comparative analyses stemming from this approach, we sought to advance the current understanding of the porcine microbiome and its association with the evolution of these animals.

Material and Methods

Animal sample collection

We selected three unrelated (no shared ancestors for 3+ generations) female adult (4-year-old) wild pigs from populations in Xingyang county of Henan province in China that were derived from similar genetic backgrounds and subjected to comparable husbandry practices. These animals were fed twice daily with a controlled diet composed of corn and soybean and supplemented with hay, which likely facilitated reductions in microbiota variability relative to wild animal populations. Animals had free access to water, and all were healthy and not subjected to any antibiotic treatments. After sacrificing these three animals, the gastrointestinal tract was removed from each animal within 30 min of death and luminal contents were collected from each of the 5 indicated segments. These contents were specifically collected from the middle of each sample, with full disinfection of the experimental tools and work area being performed between samples to prevent any microbial cross-contamination. Samples were then frozen in liquid nitrogen and stored at −80 °C. The animal care and use guidelines put forth by the Ministry of Science and Technology of China (Guidelines on Ethical Treatment of Experimental Animals (2006) No. 398) were followed for this study, with the Ethics Committee of Shangqiu Normal University having approved all experiments herein (Shang (2017) No. 168).

16S rRNA gene sequencing

A TIANamp Stool DNA Kit (DP328; Tiangen BioTech Beijing, China) was used to isolate luminal bacterial DNA based on provided directions, after which a NanoDrop One Microvolume Spectrophotometer (Thermo Fisher Scientific, DE, USA) was used to assess DNA concentrations, while 0.8% agarose gel electrophoresis was used to assess DNA quality and purity. The bacterial 16S rRNA V3–V4 region was amplified using the well-documented primer pair: 338F (5′- ACTCCTACGGGAGGCAGCA -3′) and 806R (5′-GGACTACHVGGGTWTCTAAT -3′). To facilitate multiplexed sequencing, samples were barcoded with 7-bp tags that were specific for each sample. Individual PCR reactions were composed of 5 µL Q5 reaction buffer (5 ×), 5 µL Q5 High- Fidelity GC buffer (5 ×), 0.25 µL Q5 High-Fidelity DNA Polymerase (5U/µL), 2 µL (2.5 mM) dNTPs, 1 µL (10 uM) of each primer, 2 µL template DNA, and 8.75 µL ddH2O. Thermocycler settings were: 98 °C for 2 min, then 25 cycles of 98 °C for 15 s, 55 °C for 30 s, and 72 °C for 30 s, followed by a final of 5 min at 72 °C. Phusion High-Fidelity PCR Master Mix (New England BioLabs, MA, USA) was used for PCR reactions. Next, 2% agarose gel electrophoresis was used to confirm that amplicon sizes were consistent with expectations (∼500 bp) and that samples were of good quality and purity. Then, the ∼500 bp sample band was subjected purification with a GeneJET Gel Extraction Kit (Thermo Scientific, USA) per the manufacturer’s instructions. Purified amplicons were used for library preparation. We used a Next Ultra DNA Library Prep Kit for Illumina (New England BioLabs) for library preparation per the manufacturer’s instructions, with a Qubit@ 2.0 Fluorometer (Thermo Fischer Scientific) and an Agilent Bioanalyzer 2100 machine used to evaluate library quality. Sequencing of the resultant library was performed on an Illumina MiSeq platform, generating 250-bp paired-end reads.

Sequence analysis

Sequencing data was processed using the Quantitative Insights Into Microbial Ecology (QIIME, v1.8.0) pipeline (Caporaso et al., 2010). Briefly, any sequencing reads that exactly matched barcodes were assigned as valid sequences to the corresponding samples. Any low-quality reads were then filtered to remove reads meeting the following criteria: sequences that were <150 bp long, had average Phred scores <20, contained ambiguous bases, or mononucleotide repeats >8 bp long. FLASH was used for paired-end read assembly (Magoc & Salzberg, 2011). High-quality sequences that remained after chimera detection were grouped using UCLUST into operational taxonomic units (OTUs) at the 97% sequence identity level (Edgar et al., 2011). For each OTU, a representative sequence was then selected on the basis of default parameters to facilitate BLAST-mediated taxonomic classification with the Greengenes Database (13.8 version; DeSantis et al., 2006). The abundance of the OTUs in a given sample were compiled in an OTU table. Any OTUs containing <0.001% of total sequences across all samples were discarded. To minimize the impact of variable sequencing depth among samples, we generated a rounded rarefied OTU table by taking the average of 100 evenly resampled OTU subsets under the 90% of the minimum sequencing depth for further analyses.

Bioinformatics and statistical analysis

QIIME and R packages (v3.2.0) were used to analyze sequencing data. We used the Chao1, Abundance-based coverage estimator (ACE) metric, Shannon, and Simpson alpha diversity indices, which were calculated using QIIME (Caporaso et al., 2010), to assess species diversity and complexity among samples. The richness and evenness of OTU distributions across samples were assessed based upon OTU-level ranked abundance curves. UniFrac distance metrics were used to assess variations in beta diversity among samples corresponding to structural differences in microbial community composition (Lozupone & Knight, 2005; Lozupone et al., 2007), and these metrics were visualized using nonmetric multidimensional scaling (NMDS) and unweighted pair-group method with arithmetic means (UPGMA) hierarchical clustering (Ramette, 2007). The abundance of microbes at the phylum and genus levels were compared between samples with Metastats (White, Nagarajan & Pop, 2009).

In order to identify those OTUs differing significantly among the five sample regions, we utilized the linear discriminant analysis coupled with the effect size (LEfSe) algorithm based upon relative OTU abundance (Segata et al., 2011). Briefly, this algorithm first used a non-parametric factorial Kruskal-Wallis (KW) sum-rank test to identify OTUs that were present at significantly different levels, after which pairwise Wilcoxon tests were used to assess biological consistency between groups. LDA scores were then used to yield an estimated effect size for each differentially abundant feature. PICRUSt was used to predict microbial function according to high-quality sequences (Langille et al., 2013).

Results

Sequencing data overview

We were able to successfully amplify the 16S rRNA sequences from luminal samples collected from 5 different gut sections from each of three 4-year-old wild pigs. All 15 of the resultant samples were sequenced, yielding 925,293 reads that clustered into 687,477 tags (Table S1). The sequenced raw data sets have been submitted in the NCBI databases (PRJNA575288). QIIME processing grouped these samples into 6,513 operational taxonomic units (OTUs; Table S1), and we then performed species accumulation and rank-abundance curve analyses to confirm the presence of these OTUs within each of our samples. Species accumulation and rank-abundance curve patterns were similar across samples, suggesting that most detectable bacterial species were present in most or all samples (Fig. 1A and 1B).

Figure 1 Species accumulation (A) and rank-abundance (B) curves analysis of the different gut intestinal tract samples at 97% sequences identity.

Duodenum (DU), jejunum (JE), ileum (IL), cecum (CE), and colon (CO). If the curves reach or nearly reach a plateau, it suggests that most of the species present in all samples have been observed.

Microbiota composition throughout the intestinal tract of wild pigs

To explore the composition of microbial communities in different regions of the intestinal tract of wild pigs, we initially assessed the alpha diversity of the microbiome in the 5 tested regions (Table S3). We noted significant differences in the average number of OTUs among these sections, with the highest number of OTUs being present in the duodenum (1904 ± 38) relative to the jejunum (1381 ± 97) and ileum (1377 ± 46; P <0.05), and with higher OTU numbers in cecum samples (1955 ± 138) relative to jejunum samples (1381 ± 97; P <0.05). We then calculated the ACE and Chao1 indices, which respectively measure richness/evenness and richness. Samples from the ileum and jejunum had significantly lower ACE and Chao1 values than did samples from the duodenum, cecal, and colon (P <0.05; Fig. 2A and 2B). We further used the Shannon index as an additional comparison of alpha diversity analyzing numbers of species and relative abundance within a given sample, but no significant differences in this index were detected among regions of the intestinal tract (P >0.05; Fig. 2C). Similarly, no differences in the Simpson index were detected among samples (P >0.05; Fig. 2D). We further explored how similar or variable the microbial community composition was between samples at the OTU level via NMDS and UPGMA approaches. As shown in Figs. 3A and 3B (Unweighted UniFrac), we observed significant differences in gut microbiota composition across sample regions, with the bacterial composition in samples from the duodenum, jejunum, and ileum differing significantly from cecal and colon samples, which were similar to one another (Fig. 3A). A Bray-Curtis clustering analysis revealed that the majority of gut microbes clustered into two subgroups, with the cecum and colon samples clustering separately from the ileum, duodenum, and jejunum samples via UPGMA (Fig. 3B). Together these findings clearly indicated that the composition of the gut microbiome was not uniform throughout the intestinal tract of wild pigs, with cecal and colon samples being more similar to one another than were duodenal, jejunal, and ileal samples.

Figure 2 The alpha-diversity comparisons for the duodenum (DU), jejunum (JE), ileum (IL), cecum (CE), and colon (CO).

(A) The ACE index at the sampling location (mean ±SD). (B) The Chao1 index at the sampling location (mean ±SD). (C) The Shannon’s diversity index at the sampling location (mean ±SD). (D) The Simpson index at the sampling location (mean ±SD).

Figure 3 The beta-diversity comparisons for the duodenum (DU), jejunum (JE), ileum (IL), cecum (CE), and colon (CO).

(A) Unweighted UniFrac NMDS of the microbiota. Each symbol and color denote each gut location microbiota. (B) Bray-Curtis dendrogram analyses were performed on the 16S rRNA V3–V4 region.

We additionally assessed the taxonomic distributions of the most abundant bacterial OTUs in each sample region. Based on the bacterial relative abundance of the top 17 phyla, we specified about the Firmicutes levels at each location or made it clear in which location that Firmicutes were the most prevalent phylum (52.35% of cecal microbes), followed by Actinobacteria (46.66% of jejunal microbes). In contrast, Proteobacteria levels in the cecal and colon samples were low (1.10% and 0.92%, respectively). The relative Bacteroidetes abundance in duodenal, jejunal, ileal, cecal, and colon samples was 11.60%, 0.40%, 5.10%, 9.73%, and 16.86%, respectively. We also detected Cyanobacteria (3.10%) in the duodenum, and Verrucomicrobia (1.35%) in the colon (Fig. 4A; Table S4).

Figure 4 Community composition of the gut microbiota in different intestinal segments of wild pigs at the phylum (A) and genus (B) levels, respectively.

Duodenum (DU), jejunum (JE), ileum (IL), cecum (CE), and colon (CO).

At the genus level, a total of 163 genera were identified. Bifidobacterium was most prevalent and the relative abundance was 34.87%, 31.10%, 17.21%, 22.48%, and 22.70% in the duodenum, jejunum, ileum, cecum, and colon, respectively. In addition, we also observed Lactobacillus (11.63%), Prevotella (9.97%), Unclassified_Clostridiaceae (7.62%), Unclassified_Coriobacteriaceae (6.51%), and Megasphaera (4.04%) in the duodenum. Unclassified_Coriobacteriaceae (13.52%), Psychrobacter (12.39%), Lactobacillus (11.05%), and Allobaculum (4.73%) were most prevalent in the jejunum. Psychrobacter (17.50%), Unclassified_Coriobacteriaceae (10.93%), Allobaculum (4.26%), Unclassified_Clostridiaceae (5.58%), Prevotella (4.29%), and Unclassified_Moraxellaceae (5.98%) were most prevalent in the ileum. Allobaculum (14.970%), Unclassified_Coriobacteriaceae (12.12%), Unclassified_Clostridiaceae (11.90%), Unclassified_Clostridiales (5.91%), Unclassified_Lachnospiraceae (4.96%), Unclassified_Ruminococcaceae (6.41%), and Unclassified_Bacteroidales (6.28%) were the most prevalent in the cecum, Unclassified_Coriobacteriaceae (6.80%), Allobaculum (12.80%), Unclassified_Clostridiaceae (5.21%), Unclassified_Clostridiales (4.84%), Unclassified_Lachnospiraceae (4.80%), Unclassified_Ruminococcaceae (5.21%), and Wautersiella (11.15%) were the most prevalent in the colon (Fig. 4B; Table S5). Furthermore, we found in the distribution of 33 genera of wild pig intestinal microbiota that the abundance of only two genera in the ileum is 0, the average distribution index of each genera in the ileum is the lowest, and the ileum microbiome presents a greater evenness than that at other locations. These results demonstrated the presence of many genera of bacteria across the different regions of the gut, with more uniform distribution in the ileum.

To identify the bacterial species most characteristic of the five tested gut regions, we conducted an LEfSe analysis of the taxa with LDA scores >2; this approach revealed that 28 OTUs were differentially present in the ileal, cecal, and colon samples (Fig. 5, Table S6). The relative abundance of 14 OTUs was evident in the colon samples relative to those from the ileum and cecum; these OTUs included the Verrucomicrobia, s24_27, rfp12, WCHB1_41, Verruco_5, Ruminococcus, p_75_a5, CF231, Christensenellaceae, Verrucomicrobiae, Akkermansia, Dorea, Verrucomicrobiales, and Verrucomicrobiaceae genera. We noted a relatively higher Actinomycetales abundance in the ileum relative to the other gut locations. In the cecum, 13 OTUs were present with higher abundance than that in other gut regions, with one representative OTU among these 13 being Ruminococcaceae (Fig. 5).

Figure 5 Bacterial taxa differentially represented in ileum (IL), cecum (CE), and colon (CO) gut locations in wild pigs identified by LEFSe using an LDA score threshold of >2.0.

Figure 6 Predicted functional of the gut microbiota in the duodenum (DU), jejunum (JE), ileum (IL), cecum (CE), and colon (CO).

The vertical columns represent groups, and the horizontal rows depict metabolic pathways. The color coding is based on row z-scores.

Functional analysis of the gut microbiota along the intestine tract

We next used PICRUSt to develop an understanding of the metagenomic activity of the identified bacteria across our samples, as such functional assessments of the microbiome may offer more meaningful insights into the spatial distinctions in metabolic activity across the length of the intestinal tract. These metagenomic inferences were made based upon available annotations for the detected OTUs in this study. The genes identified through this metagenomic analysis were then aligned to the KEGG database to gain functional insights. Through this approach, we identified 5945 KEGG genes (Table S9) that were assigned to 289 pathways (Table S8). We then assessed the relative abundance of these pathways among samples from different gut regions (P <0.05; Fig. 6), identifying 15 significantly differentially enriched pathways (Fig. 7). A total of 10 pathways were significantly enriched in ileal samples (Caprolactam_degradation, Glutathione_metabolism, Benzoate_degradation, Cytochrome_P450, Drug_metabolism___cytochrome_P450, Metabolism_of_xenobiotics_by_cytochrome_P450, Alzheimer_disease, Parkinson_disease, Cardiac_muscle_contraction, Apoptosis), while 3 were more enriched in cecal samples (Methane_metabolism, Epithelial_cell_signaling_in_Helicobacter_pylori_infection, Germination), and 2 were more enriched in jejunal samples (Tyrosine_metabolism, Chloroalkane_and_chloroalkene_degradation. The pathways that were enriched in ileum samples were associated with glycerophospholipid metabolism (ko02029), diterpenoid biosynthesis (ko09686), and bacterial chemotaxis (ko02030). The pathways enriched in cecum and colon samples were linked with steroid hormone biosynthesis (ko03088), the pentose phosphate pathway (ko00615), and arginine and proline metabolism (ko01990, ko01992). The pathways enriched in jejunum samples were associated with fatty acid metabolism (ko01897 and ko04924), ABC transporters (ko02006, ko02008), the biosynthesis of unsaturated fatty acids (ko02050), and cardiovascular diseases (ko05410; Fig. 8; Table S9). Together these findings suggest that the functional metabolic activity of the microbiome varies over the length of the intestinal tract, with certain pathways being preferentially engaged in a spatially-defined manner.

Figure 7 Predicted functional differentially of the bacterial genus represented in the jejunum (JE), ileum (IL), and cecum (CE) gut locations in wild pigs identified by LEFSe using an LDA score threshold of >2.0.

Figure 8 Heatmap clustered by the KEGG pathway showing different enrichments in the duodenum (DU), jejunum (JE), ileum (IL), cecum (CE), and colon (CO) of wild pigs.

The vertical columns represent groups, and the horizontal rows depict metabolic pathways. The color coding is based on row z-scores.

Discussion

High throughput sequencing analyses have facilitated rapid advances in our understanding of the intestinal microbiome over the last decade, facilitating both functional and compositional analyses of this complex microbial community. The specific microbial composition and evolution of the intestinal tracts of wild pigs, however, remain poorly characterized. As such, we utilized 16S rRNA sequencing to survey the composition, distribution, and function of intestinal microbiome across different regions of the digestive tract in Chinese wild pigs. In addition, previous studies have not investigated the link between the gut microbiome and porcine evolution. Hence, in this study, we also inferred the relationship between variations in the porcine gut microbiota compositional and pig evolution, providing novel insights into the evolution of pigs.

The primary findings of this study centered on exploring the structural diversity of the intestinal microbiome in wild pigs. We found that the bacterial composition of all tested samples was dominated by Firmicutes (30.45%-52.35%), Actinobacteria (31.22%-46.66%), Proteobacteria (0.92%-32.39%), and Bacteroidetes (0.40%-16.86%) at the phylum level (Fig. 4A and Table S4). This was in contrast to previous studies of the intestinal microbiome of domestic pigs (Crespo-Piazuelo et al., 2018; Gao et al., 2019; He et al., 2016; Ivarsson et al., 2014; Kraler et al., 2016; Liu et al., 2012; Mach et al., 2015; Quan et al., 2018; Quan et al., 2019; Ramayo-Caldas et al., 2016; Slifierz, Friendship & Weese, 2015; Xiao et al., 2018; Yang et al., 2016), which found the core microbiome in these animals to be dominated by Firmicutes and Bacteroidetes. In contrast, we found Actinobacteria to be the second most dominant bacterial phyla in the intestines of wild pigs. Thus, in this study, besides the fact that Firmicutes dominated both wild pigs and domestic pigs, Actinobacteria in wild pigs dominated and Bacteroidetes in domesticated pigs predominated. Bacteroidetes are naturally competent Gram-negative bacteria (Mell & Redfield, 2014) and can also degrade bacterial exopolysaccharides in animal intestines (Lammerts van Bueren et al., 2015). An increasingly robust body of evidence has shown that Actinobacteria species are abundant in the intestines of animals, wherein they can produce key antibiotics, immunomodulatory compounds, and metabolites that are vital to host health and homeostasis (Matsui et al., 2012). Of note, domestic pigs are domesticated by wild pigs. During the evolution of pigs, the intestinal microbes in pigs also coevolved. From wild pig’s Actinobacteria to the domestic pig Bacteroidetes phyla predominance, not only related to the pig evolution but also related to pig genetics, natural selection, environment, and feeding system. Overall, the well-documented role of intestinal Actinobacteria species as promoters of antibacterial resistance indicates that the higher levels of these bacteria in the intestines of wild pigs could correspond to improved disease resistance and roughage nutrient resistance relative to domestic pigs.

We found that Bifidobacterium and Allobaculum were the most abundant genera of bacteria in most of the tested wild pig intestinal samples, corresponding to 2.08–34.87% of total bacteria in these samples (Fig. 4B; Table S5). However, the relative abundance of particular bacterial taxa varied substantially between different sites in the intestines. The most noticeable locations were found in the duodenum and jejunum, where Bifidobacterium relative abundance was significantly higher than in the ileum, cecum, and colon (31.10%–34.87% vs 17.21%–27.40%; Fig. 4B; Table S5). Allobaculum comprised over 10% of the cecal and colonic bacteria in these wild pigs, being similarly dominant to Clostridia, Psychrobacter, and Lactobacillius genera in this region of the intestine. Past studies have yielded significant variations in the genera-level intestinal microbiome composition of pigs. For example, Yang et al. (2016) found Prevotella, Lactobacillus, and Treponema to be the most abundant genera in Duroc pigs, while Xiao et al. (2017) found Prevotella, Streptococcus, and SMB53 to be the most abundant genera in Hampshire pigs, with Clostridium, SMB53, and Streptococcus being the most abundant in Landrace and Yorkshire pigs. This suggests that there are significant differences in microbiome composition at the genus level among pigs, possibly owing to differences in age, breed, feed composition, or husbandry practices.

In addition, Bifidobacterium species are a major probiotic species in humans, playing roles in digestion, nutrient absorption and metabolism, and disease resistance owing to their ability to maintain the integrity of the mucosal barrier in the intestines (Furusawa et al., 2013). Bifidobacterium have, for example, been shown to help prevent rotavirus enteritis (Rigo-Adrover et al., 2017) and necrotizing enterocolitis in premature rats (Satoh et al., 2016; Wu et al., 2013), and to bolster immune function and inflammation in weaning rats with colitis (Izumi et al., 2015). Additional research has further found that Bifidobacterium can help prevent cardiac damage (Sadeghzadeh et al., 2017) and can influence the development of metabolic syndrome (Bordoni et al., 2013; Kim et al., 2017; Plaza-Díaz et al., 2017a; Plaza-Díaz et al., 2017b; Plaza-Diaz et al., 2014; Zhu et al., 2018). Short-chain fatty acid (SCFA) producing bacterial genera, including Bifidobacterium and Allobaculum, have been found to provide beneficial effects to hosts through these SCFAs, reducing inflammation and promoting colonic health. Similarly, Allobaculum has been found to be inversely correlated with adiposity, and the abundance of these microbes correspondingly increased in C57BL/6 mice fed a low-fat diet relative to those mice fed a high-fat diet (Baldwin et al., 2016). We, therefore, hypothesize that the high levels of Bifidobacterium and Allobaculum in the intestines of the microbiome may help to promote nutrient absorption and disease resistance in wild pigs. Moreover, Ushida et al. (2016) examined the impact of domestication and modern feeding practices on the intestinal microbiome composition in Suidae pigs through metagenomic analyses, revealing higher relative Bifidobacterium abundance in the gut of wild pigs relative to domestic pigs. They further suggested that domestication and/or modern feeding practices may have led to the relative dominance of Lactobacillus that they observed in the guts of domesticated pigs, as these species were only in the top 20 genera of bacteria present in the guts of wild pigs. The exact factors driving this change remain uncertain, and any exploration thereof necessitates the genomic analyses of Suidae-associated Bifidobacterium species in an effort to identify genetic changes in these bacteria that may have allowed them to adapt to growth in domesticated pigs subjected to modern feeding practices. To that end, Tsuchida et al. (2017) analyzed 7 strains of Sus-associated Bifidobacterium via genomic alignment (3 from domestic pigs and 4 from free-range wild pigs); they found that the bacterial isolates from wild pigs expressed enzymes associated with fiber degradation, and the bacterial isolated from domesticated animals expressed functional tetracycline-resistant genes. The expression of functional tetracycline resistance genes from intestinal bacterial of domestic animals could be related to the routine use of antibiotics to treat these animals during growth and development. Overall, these authors observed clear differences in the gut microbiome of wild pigs relative to their domestic counterparts (Ushida et al., 2016; Tsuchida et al., 2017). Together these previous results suggest that over the course of porcine evolution and domestication, significant environmental and nutrient source changes are, together with artificial selection, likely to result in a divergence in intestinal microbiome composition in these animals such that modern domestic pigs bear a microbial cohort associated with rapid growth but poor disease resistance relative to wild pigs.

In the final section of our study, we conducted a metagenomic analysis of the functional capacity of the intestinal microbiome in wild pigs, leading us to identify 15 significantly differentially enriched pathways across spatial regions within the intestine (P <0.05; Fig. 6). In total 2 pathways were more enriched in jejunal samples, 10 were more enriched in ileal samples, and 3 were more enriched in cecal samples (Fig. 7). The main metagenomic activities of the jejunal microbiome in these wild pigs were related to carbohydrate metabolism via glycolysis and/or gluconeogenesis. In contrast, metagenomic functions in the ileum were more closely associated with fatty acid and pyruvate metabolism and xylene degradation, likely owing to the abundance of Clostridiales in this region (Niu et al., 2015). The predominant metagenomic functions evident in the cecum of these wild pigs were related to carbohydrate and lipid metabolism, whereas protein metabolism was enriched in colonic samples. We further found a preferential abundance of methane metabolism in the cecum, which is the predominant site of methane generation, the relative abundance of which increases nearer to the ends of the intestines. Production of SCFAs following dietary polysaccharides fermentation have been previously shown to improve intestinal absorptive capacity and feed efficiency in pigs (Yang et al., 2017; Pryde et al., 2002). We also found different pathways enriched in ileal, cecum, colon, jejunal samples (Fig. 8; Table S9). These metagenomic analysis results thus suggest that the gut microbiome exhibits distinct functional and spatial organization that helps to facilitate the rapid degradation and utilization of diverse nutrient sources by local bacterial species that are able to proliferate and maintain gut homeostasis. Further work, however, will be needed to confirm our results which are largely predictive in nature.

Conclusions

In summary, we observed significant differences in the microbial community structures in different regions of the intestinal tract of wild pigs. Of note, we found the Actinobacteria microbiota of these wild pigs to be highly distinct from those of domesticated pigs subjected to domestication, environmental and nutrient source changes, and artificial selection. While Bifidobacterium and Allobaculum of these forms of bacteria are present within the intestines of wild pigs, their densities are also much higher than that in domesticated animals. This indicates that in the process of pig evolution, the intestinal microbes also evolved, and changes in the intestinal microbial diversity are one of the main driving forces for the evolution of pigs. Functional analyses of the intestinal microbiome in these wild pigs revealed that there were a number of distinct metabolic pathways and spatial structures within this system. Overall, these results offer unique insights that would help to advance the current understanding of how the intestinal microbes interact with the host and affect the evolution of pigs.

Supplemental Information

Table S1 Overview of high-throughput sequencing data in different gut location samples from wild pigs

Click here for additional data file.

Table S2 Operational taxonomic units (OTUs) were obtained among the five gut sections in wild pigs

Click here for additional data file.

Table S3 OTU numbers of the different gut intestinal samples at different taxonomy levels

Click here for additional data file.

Table S4 Microbial composition of the five gut intestinal in wild pigs at the phylum level

Click here for additional data file.

Table S5 Microbial composition of the five gut intestinal in wild pigs at the genus level

Click here for additional data file.

Table S6 The analysis results of the 5 gut locations bacterial taxa differentially in wild pigs by LEFSe

Click here for additional data file.

Table S7 Predicted KEEG genes of the gut microbiota in five gut segments in wild pigs

Click here for additional data file.

Table S8 Predicted KEEG pathway of the microbiota in five gut segments in wild pigs

Click here for additional data file.

Table S9 The results of pathway enrichment based on OTUs to the annotated genes

Click here for additional data file.

Supplemental Information 1 Y6DU1_R1 raw data exported from the Illumina MiSeq platform and applied for data analyses

Click here for additional data file.

Supplemental Information 2 Y6DU1_R2 raw data exported from the Illumina MiSeq platform and applied for data analyses

Click here for additional data file.

Supplemental Information 3 Y7DU1_R1 raw data exported from the Illumina MiSeq platform and applied for data analyses

Click here for additional data file.

Supplemental Information 4 Y7DU1_R2 raw data exported from the Illumina MiSeq platform and applied for data analyses

Click here for additional data file.

Supplemental Information 5 Y8DU1_R1 raw data exported from the Illumina MiSeq platform and applied for data analyses

Click here for additional data file.

Supplemental Information 6 Y8DU1_R2 raw data exported from the Illumina MiSeq platform and applied for data analyses

Click here for additional data file.

Supplemental Information 7 Y6JE2_R1 raw data exported from the Illumina MiSeq platform and applied for data analyses

Click here for additional data file.

Supplemental Information 8 Y6JE2_R2 raw data exported from the Illumina MiSeq platform and applied for data analyses

Click here for additional data file.

Supplemental Information 9 Y7JE2_R1 raw data exported from the Illumina MiSeq platform and applied for data analyses

Click here for additional data file.

Supplemental Information 10 Y7JE2_R2 raw data exported from the Illumina MiSeq platform and applied for data analyses

Click here for additional data file.

Supplemental Information 11 Y8JE2_R1 raw data exported from the Illumina MiSeq platform and applied for data analyses

Click here for additional data file.

Supplemental Information 12 Y8JE2_R2 raw data exported from the Illumina MiSeq platform and applied for data analyses

Click here for additional data file.

Supplemental Information 13 Y6IL3_R1 raw data exported from the Illumina MiSeq platform and applied for data analyses

Click here for additional data file.

Supplemental Information 14 Y6IL3_R2 raw data exported from the Illumina MiSeq platform and applied for data analyses

Click here for additional data file.

Supplemental Information 15 Y7IL3_R1 raw data exported from the Illumina MiSeq platform and applied for data analyses

Click here for additional data file.

Supplemental Information 16 Y7IL3_R2 raw data exported from the Illumina MiSeq platform and applied for data analyses

Click here for additional data file.

Supplemental Information 17 Y8IL3_R1 raw data exported from the Illumina MiSeq platform and applied for data analyses

Click here for additional data file.

Supplemental Information 18 Y8IL3_R2 raw data exported from the Illumina MiSeq platform and applied for data analyses

Click here for additional data file.

Supplemental Information 19 Y6CE4_R1 raw data exported from the Illumina MiSeq platform and applied for data analyses

Click here for additional data file.

Supplemental Information 20 Y6CE4_R2 raw data exported from the Illumina MiSeq platform and applied for data analyses

Click here for additional data file.

Supplemental Information 21 Y7CE4_R1 raw data exported from the Illumina MiSeq platform and applied for data analyses

Click here for additional data file.

Supplemental Information 22 Y7CE4_R2 raw data exported from the Illumina MiSeq platform and applied for data analyses

Click here for additional data file.

Supplemental Information 23 Y6CO5_R1 raw data exported from the Illumina MiSeq platform and applied for data analyses

Click here for additional data file.

Supplemental Information 24 Y6CO5_R2 raw data exported from the Illumina MiSeq platform and applied for data analyses

Click here for additional data file.

Supplemental Information 25 Y7CO5_R1 raw data exported from the Illumina MiSeq platform and applied for data analyses

Click here for additional data file.

Supplemental Information 26 Y7CO5_R2 raw data exported from the Illumina MiSeq platform and applied for data analyses

Click here for additional data file.

Supplemental Information 27 Y8CO5_R1 raw data exported from the Illumina MiSeq platform and applied for data analyses

Click here for additional data file.

Supplemental Information 28 Y8CO5_R2 raw data exported from the Illumina MiSeq platform and applied for data analyses

Click here for additional data file.

Supplemental Information 29 Y8CE4_R1 raw data exported from the Illumina MiSeq platform and applied for data analyses

Click here for additional data file.

Supplemental Information 30 Y8CE4_R2 raw data exported from the Illumina MiSeq platform and applied for data analyses

Click here for additional data file.

Additional Information and Declarations

Competing Interests

Author Contributions

Animal Ethics

Data Availability

The authors declare there are no competing interests.

Guangli Yang conceived and designed the experiments, analyzed the data, prepared figures and/or tables, authored or reviewed drafts of the paper, and approved the final draft.

Chuanxin Shi and Shuhong Zhang analyzed the data, prepared figures and/or tables, authored or reviewed drafts of the paper, and approved the final draft.

Yan Liu, Zhiqiang Li, Fengyi Gao and Yanyan Cui performed the experiments, prepared figures and/or tables, authored or reviewed drafts of the paper, and approved the final draft.

Yongfeng Yan and Ming Li conceived and designed the experiments, prepared figures and/or tables, authored or reviewed drafts of the paper, and approved the final draft.

The following information was supplied relating to ethical approvals (i.e., approving body and any reference numbers):

The animal care and use guidelines put forth by the Ministry of Science and Technology of China (Guidelines on Ethical Treatment of Experimental Animals (2006) No. 398) were followed for this study, with the Ethics Committee of Shangqiu Normal University having approved all experiments herein (Shang (2017) No. 168).

The following information was supplied regarding data availability:

The sequenced raw data sets are available at GenBank: PRJNA575288.

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
