# Peer review of "Characterization of the bacterial microbiota composition and evolution at different intestinal tract in wild pigs (Sus scrofa ussuricus)"

_PeerJ, doi:10.7717/peerj.9124_

## Round 0.1 · original submission · Major Revisions

After careful evaluation, I recommend you make extensive changes in the manuscript as suggested by three reviewers.

·

Basic reporting

English used in the article is at a satisfactory level, the whole manuscript was written technically correct. First part of the paper explains in a clear way recent state of knowledge in the field of Sus scrofa ussuricus microbiomes. Notwithstanding, the authors did not avoid some minor misprints in this part:
• Line 25 – “behavior”
• Line 86 – “fecal”
• Line 95 – “micobiota”
All of presented research to show the background of the study is well referenced by articles published in reputable journals. Introduction leads us through the microbiome research across all vertebrate species. Text is clearly divided into separate sections, which make it easier to read and understand.
According to figures supplied:
• Figure 1 – these images are readable only when opened as a separate file. Fonts in axes descriptions are too small to be readable. Legend with rank-abundance on the right is completely invisible and makes it unclear without zooming in.
• Figure 2 - description of the figure is cut in the middle. There is no information about panels C and D.
• Figure 3 - description of the figure is cut in the middle. Panels do not have assigned letters (A and B) which are referenced to in the part of the explanation.
• Figure 4 – it is hard to estimate percent values for specific groups with this type of scale on the y-axis. The authors should consider adding visible values to the diagram separately.
• Figure 5 – In the description the authors mentioned that diagram presents bacterial genus comparison while in the vertical row bacterial orders and families are placed. Position of Spirochaetes is doubled.
• Figure 6 – description cut in the middle.
• Figure 7 – description is missing and not adequate to presented diagram.
• Figure 8 –“KEEG” is misspelled, description cut off in the middle. Vertical row without any commentary is completely unclear.
According to raw data supplied:
• Raw sequences were uploaded correctly to PeerJ server and are possible to reanalyse.
• It was impossible to access raw sequences via GeneBank with the attached number and submitted link. It shows that “The requested page does not exist”.

Experimental design

Research conducted by the authors meets Aims and Scope of PeerJ. Research question is clearly described in the methods section.
Line 106 - Experiment was designed to use only three animals, which may not be enough to draw far-reaching conclusions. What is more, only female individuals were used. Can the authors provide more information about the process of choosing the quantity of animals and their gender?
In lines 106 – 117 the authors describe sample preparation process. Luminal contents seem to be an appropriate material for microbiome research in general. In this specific study, to compare different parts of the intestinal tract, this approach can be questionable. Obtained luminal content contains DNA not only from the specific point of the tract that it was taken from. It consists of remains of undigested food from the previous parts of the gastrointestinal tract as well as dead bacterial cells which were living along the digestive system. Isolation of DNA and further 16S rRNA gene sequencing does not provide any distinction between identified living and dead bacterial fractions. It can lead to false positive results as sequencing can show mixed fractions from different regions of the gastrointestinal tract because of dead bacterial cells accumulation in luminal contents along the entire length of the intestine. Can the authors explain why they used contents instead of a different type of material e.g. specimens?
In line 123 the authors mentioned which 16S rRNA regions were amplified to conduct the study. Why did the authors choose only V3-V4 for this analysis? Can the authors provide any reference materials about previous studies with this type of investigation?
In line 135 the authors start to describe their library preparation process, but the information about final DNA input is missing.
In line 141 the authors showed that an outdated, no longer supported version of QIIME was used. Was there any reason to not use the most up to date QIIME2 compilation?
Sequence analysis was performed according to the previous well referenced pipeline. Unfortunately, in line 150 there is no information about Greengenes Database version used in analyses. Can authors provide this information?

Validity of the findings

Line 194 – As mentioned before, higher OTU number can be a result of accumulation of bacteria from the preceding sections of the gastrointestinal tract.
Line 265 – the authors mentioned that 3 more enriched pathways were detected in cecal samples, among others – “Epithelial_cell_signaling_in_Helicobacter_pylori_infection”. How was it possible given that H. pylori is not able to infect pigs naturally. I did not observe any H. pylori presence in the samples. Can the authors provide any information about this specific point of analysis? Definitely the part with functional analysis of the gut microbiota along the intestine tract should be extensively explained in the discussion section.
Conclusions were written clearly, according to obtained results. In 399-401 line it is suggested that “This indicates that in the process of pig evolution, the intestinal microbes in vivo also evolved, and changes in the intestinal microbial diversity is also the main driving force for the evolution of pigs.”. There are not enough samples to draw such conclusion. A group of three individuals, all representing only one gender, is too small to extrapolate results for the whole species. More research aimed at different groups should be conducted.

Additional comments

To sum up, this submission is an examples of a well-prepared study designed with the use of latest technologies that can help extend the knowledge about the role of microbiome in Sus scrofa ussuricus . Nevertheless, I had some questions about the research methods that I mentioned above. Most importantly, I would like to know why the authors used luminal contents instead of local specimens. One of the issues is also to properly share the raw data from sequencing via GenBank. The link is misleading, hence the upload may need additional action.

Thank you for reading my review. I hope it will help you make your manuscript even better.

Reviewer 2 ·

Basic reporting

no comment

Experimental design

Article is technically sound.

Validity of the findings

Finding and presentation of result is satisfactory.

Additional comments

Yang et al. have presented an article "Characterization of the bacterial microbiota composition and
evolution at different intestinal tract in wild pigs (Sus scrofa
ussuricus)" where V3-V4 region of 16 S rRNA is used for preparation of metagenomic library. Upon analysis it is found that 6513 OTU of 17 phyla that include 163 genera of bacteria resided in to intestine if wild pig. Among all Actinobateria were found to be predominant in test condition. Authors emphasized that during course of evolution intestinal bacterial community changes and it happened in pig too. Article is well written and I would recommend this to be published in Peer J.

Reviewer 3 ·

Basic reporting

It is highly recommended to have the manuscript revised by an English speaker. There are many run-on sentences that are difficult to understand. Certain statements need be supported by either data analysis or previously published papers, so please cite appropriate papers. Legends of some figures are incomplete or need to be rewritten. The experimental design and results are not good enough to make the conclusions in the manuscript.

Experimental design

The title of the paper was characterizing the bacterial microbiota composition and evolution in wild pigs, but more than half of the manuscript was comparing wild pig microbiome with domesticated pigs which is missing in the experimental design. The sample size is small. Also, the wild pigs were housed in specific area and fed with general feeds (such as corn/hay) and water before sample collection, which could impact the gut microbiome significantly. These are major flaws in the experimental design.

Validity of the findings

The paper provides good information about the sequencing quality. Advanced statistical tools such as LEfSe were used for data analysis. Significant differences related to microbiome composition and metabolic pathways were achieved from the five locations of intestinal tract in wild pigs, but due to the flaws in experimental design, the findings were not trustable.

Additional comments

Abstract:
1. Line 26. Remove “however,”.
2. Line 28. Only wild pigs were mentioned. Need sample details of domestic pigs as well.
3. Line 29. Delete “in these samples”
4. Line 36. “in vivo” is redundant.
5. Line 37. Conclusion related to intestinal microbial diversity cannot be supported by demonstrated results in the abstract.
6. Line 39. Boars normally refer to male pigs and sows refer to adult female pigs. Please correct.


Introduction
1. Line 55-58. Please rewrite this sentence.
2. Line 65-66. Rewrite this sentence. A recommendation here: “Domestic pigs are now distinct from their wild counterparts in many ways such as phenotype, as evidenced by coat color differences.”
3. Line 85-89. There is no logical connection in the big sentence. Please reword.
4. Line 92-96. Cannot understand what the author tried to explain. Please reword.
5. Line 107. The “female adult wild boars” needs to be corrected. The author needs differentiate boars and sows.
Materials and methods
1. How long have you raised these three wild pigs with feeding corn soybean diets before sacrifice? They are no longer wild pigs once regular diet was introduced.
2. Line 157. Spell out the whole words of “ACE” as the first appearance in the paper.

Results
1. Line 191-194. Please clearly indicate the alpha indexes for each GI tract location with real numbers especially for OTU number. The p-values don’t provide enough information.
2. The legend of Figure 2 is not complete. The x-axis labels of each location were not showing correctly in Figure 2 C and D.
3. Line 191. It is better to present the observed OTU number/averaged OTU with other alpha indexes (Chao1, Shannon, Simpson and ACE) in the boxplot instead of showing them in a supplementary table. The OTU numbers are just as important as other indexes.
4. In Figure 3, text in the legend “group” is not in an appropriate order. Labels “(A)” and “(B)” should be shown on corners of both sub-pictures in Figure 3.
5. Line 214. How many phyla were detected in the pig gut and why do you only focus on the top 17? How did you pick this number? The abstract Line 30 depicted 17 phyla have been observed.
6. Line 215. Rewrite the sentence “We found that Firmicutes were the most prevalent phylotype (52.35% of cecal microbes.......”. Be specific about the Firmicutes levels at each location or make it clear in which location that Firmicutes are the most prevalent phylum.
7. Line 238. The statement “with more uniform distribution in the ileum” could not be supported by the composition profile directly. I assume you were trying to describe the ileum microbiome has a greater evenness than it at other locations, and that should be indicated by evenness index.

8. Line 240-249. This is a three-way (three locations) comparison. Any statement saying specific strain level in one location being higher than both other two locations has to be supported by bar-charts or tables showing the relative abundances in all three groups. You can provide this data in the supplementary file.
9. Line 266. Legend of Fig 7 requires correction. This chart is showing pathways/functions instead of genus levels.

Discussion
1. The first paragraph in the discussion needs to be refined. Most information has been depicted in the introduction.
2. Line 312. Please add a citation to the sentence “During the evolution of pigs, the intestinal microbes in pigs has also coevolved.”
3. Line 313-315. How did you get the statement “From wild pig’s Actinobacteria to the domestic pig Bacteroidetes …” Please clarify or cite the source of papers.
4. Line 318. Please add citations to the statement in the last sentence of this paragraph.
5. Line 321. Delete “overall”.
6. Line 323-325. Rewrite the sentence. May need delete the “<” in the bracket.
7. Line 326. Replace “>” with some words such as “over” and “greater than”.
8. Line 362-366. Please separate the big sentence into 2 or 3 to make the statement clear. Try avoiding using big sentence when you are not comfortable with the language.
9. Line 366. It is not appropriate to make this conclusion based on the current data. The author only sequenced the gut microbiome from three “wild” pigs and compare it with a bunch of previously published papers without truly downloading/analyzing any data of domestic pigs with similar age, gender or locations. There is an issue in terms of experimental design and data interpretation for comparison purposes between domestic and wild pigs.
10. Line 367-371. Please cite the paper that relates to disease resistance in wild and domestic pigs.

Conclusions
1. The conclusion cannot be supported by the experimental design because of two main reasons. First, the wild pigs were fed with specific diet (corn, soybean and hay) before sample collection, so they have been raised the same way as domesticated pigs. Diet and environment can have very significant effects on the gut microbiome and should not be ignored. Second, there is no comparable group of domesticated pigs with similar age, gender and other related background in the study so there is no way to compare the “wild” pigs with domesticated pigs directly.

---

## Round 0.2 · Minor Revisions

Please provide the necessary changes as suggested by the reviewer.

·

Basic reporting

Clear English was used throughout the article. Raw data has been uploaded and was available for reanalyse. The paper contains a description of the work that has been done independently, in a field that has been poorly researched so far.
In the first part of article, authors describes clearly why described work was necessary to develop and introduce the reader to the current state of knowledge. Article structure is good. Figures are well described in accordance with generally accepted criteria and complemented by supplementary data giving a full picture of the information presented. Nevertheless more findings about next generation sequencing methods would be appropriate in this section. Research question was appropriately formulated.
References were added correctly.

Experimental design

Original primary research was well proven and described, however the experiment was carried out with a relatively small number of unique samples. Methods have been described in sufficient detail to replicate according to generally used standards. Although the software for bioinformatic analysis has been used correctly, in the future I recommend using databases that are further developed (for ex Silva). 7 years is an extremely long time for the development of bioinformation databases. Statistical analysis was made correctly with widely used and described tools. Unfortunately, authors did not attach names and versions of used R packages, which would be more important to reproduce the analysis Manuscript contains necessary information about databases used in next-generation sequencing data analyse. Sequencing technical details have been correctly provided in supplementary files.

Validity of the findings

In the discussion section authors have correctly placed the subject of their research in a broad context. They carefully explained point by point obtained results and discuss its importance to the field. Nevertheless some facts about comparisons to human or other animals are confusing for reader. Conclusions were drawn with due caution. Findings described in the study are original and unique but more research is still needed to establish this, and the authors are aware of this right.

Additional comments

The authors have significantly raised the level of work by following the comments to the previous version of the manuscript. Explanations in the rebuttal letter were reasonable and clear. Errors from previous versions were corrected and contentious issues were clarified in the text. Scientific relevance of the article has been enhanced by the suggested amendments

Reviewer 2 ·

Basic reporting

Satisfactory

Experimental design

Satisfactory

Validity of the findings

Satisfactory

Additional comments

Revised manuscript should be accepted for publication.

---

## Round 0.3 · accepted · Accept

After clearing final doubts, this article is now considered for the publication.